# Ethical acceptability of human challenge trials: Consultation with the US public and with research personnel

**James William Benjamin Elsey**[1]*, **David Manheim**[2], **Abigail Marsh**[3], **Virginia Schmit**[4], **David Moss**[1]

1 Rethink Priorities, San Francisco, California, United States of America, 2 Technion, Israel Institute of Technology, Haifa, Israel, 3 Department of Psychology, Georgetown University, Washington, DC, United States of America, 4 1Day Sooner Research Team, Delaware, United States of America

* jamie@rethinkpriorities.org

**Data Availability Statement:** Data is available to download from the Open Science Framework at the following link: https://osf.io/u2ev9/?view_only=c971b7aa71ed4675ae34856797d31bfd.

## Abstract

Human challenge trials (HCTs) may accelerate the development of treatments and vaccines, and deliver novel insights into the course and consequences of infection. However, HCTs are contentious because they involve purposely exposing volunteers to infection. Consultation with the public and other stakeholders is essential for understanding how HCTs can be most ethically and acceptably pursued. Previous research has found public support for COVID-19 HCTs, but little research has considered public attitudes towards HCTs in principle and the various factors making a trial more or less acceptable. Empirical data on the attitudes of research personnel is also missing. We generated an online survey covering overarching support/opposition towards HCTs, as well as factors of importance for deciding whether or not an HCT is ethically acceptable. Our sample of the US public represents the responses of 1500 participants sampled via *Prolific*, poststratified to be representative of the general US adult population. We additionally collected a convenience sample of 33 research personnel engaged in phase III clinical trials for infectious diseases. Estimates for the US public suggest substantial support for using HCTs to develop new vaccines, new treatments, and knowledge about diseases, with similarly high support among research personnel. The most important factors in determining acceptability of an HCT were the risk to participants and their comprehension of this risk. The general public, in particular, appear relatively unconcerned about participants' motivations, and favor higher payment in accordance with risk. This study adds to a growing body of public consultation surrounding HCTs, demonstrating high levels of support for their use in principle–not just in relation to COVID-19. The importance attributed to various ethically-relevant factors can help in designing HCTs with high public acceptance.

## Introduction

Infectious diseases continue to represent a substantial proportion of the global burden of disease [1]. Beyond this ongoing burden, preparedness for future pandemics has risen as a global

**Funding:** This project was supported by an Astral Codex Ten Grant to 1DaySooner, executed through the Center for Effective Altruism.

**Competing interests:** This work was supported by an ACX Grant to 1Day Sooner, a 501(c)(3) non-profit organization that advocates for people who participate and want to participate in high-impact medical studies, including HCTs. Rethink Priorities was contracted by 1Day Sooner to execute this research project and collected and analyzed data independently of 1Day Sooner. This does not alter our adherence to PLOS ONE policies on sharing data and materials.

priority [2], spurred on by the medical, societal, and psychological burden resulting from the COVID-19 pandemic. As a part of efforts to combat infectious diseases, several researchers have raised the possibility that a specific type of medical research, known as Human Challenge Trials (HCTs), might accelerate the production of vaccines and treatments, as well as provide unique insights into disease progression and biomarkers of infection or immunity [3–6]. HCTs, also known as Controlled Human Infection Models (CHIMs), are a type of medical experiment in which volunteers are deliberately infected with a pathogen. By guaranteeing pathogen exposure, such experiments may allow for inferences regarding the efficacy of vaccines and treatments to be made more rapidly, with fewer participants, and at lower cost, than typical trials [7]. HCTs have already been used to help understand infection from a host of pathogens, and to aid in the development of treatments and vaccines for them. Examples include malaria, influenza, dengue, and cholera [8, 9]. In 2021, the UK Government approved the first HCTs involving SARS-CoV-2 [10, 11].

Although the COVID-19 pandemic has prompted increased attention to HCTs, such trials are not just an emergency option, nor one reserved for when there are no vaccines or treatments available. For example, the efficacy of existing vaccines is threatened when novel strains of a virus emerge. There may also be new vaccine candidates with greater efficacy, ease of transport and storage, more acceptable means of administration, or lower prices, than currently-approved vaccines. However, the presence of existing vaccines and natural exposure pose challenges for Phase III efficacy trials [12]: high rates of vaccination or natural immunity present a challenging immunological landscape, in which it may be difficult to isolate the impact of the tested vaccine. Such trials might also need to demonstrate non-inferiority versus existing vaccines, requiring even larger study cohorts than a placebo-controlled design. Moreover, studies of candidate treatments and prophylactics can be hampered by factors such as difficulties in timing of interventions relative to pathogen exposure, heterogeneity of viral exposure (e.g., in terms of variants and viral load), and poor knowledge of baseline health among patients who come for treatment [12]. Such issues similarly confound the search for biomarkers of immunity, and the general understanding of the progression of disease from infection. It has been argued that HCTs are uniquely positioned to address these issues [12, 13].

However, HCTs are ethically contentious: they involve purposely exposing otherwise healthy individuals to a potentially dangerous or even deadly pathogen [4, 14, 15]. This may be especially problematic with regards to diseases for which 'rescue treatments' that could save a volunteer from the most serious consequences of infection have yet to be firmly established [14]. Phenomena such as 'long-COVID' also highlight that there are always unknowns in terms of risks people are being exposed to. It has additionally been suggested that public trust of vaccines and the medical establishment more generally could be damaged if a serious adverse event occurred in such a trial [6]. This could hinder broader efforts at controlling a pandemic, or reduce trust in vaccines and the medical community more generally. Public attitudes towards HCTs are therefore a key component of their ethical viability [5, 16]. If the public finds such trials to be unacceptable or untrustworthy, they may be less likely to take up those treatments and vaccines that could result from them, or even actively push against them. This would preclude realization of the benefits that trial participants have put themselves at risk to produce. At present, few studies have investigated public perception of HCTs.

One focus-group study conducted in April 2020 with 57 adult (aged 20–40) respondents from the UK suggested favorable impressions of HCTs as a means of combating COVID-19 specifically [16]. Respondents in this study indicated that they thought such trials could be ethically acceptable, with many indicating that they would be willing to take part in such a trial with appropriate compensation and precautions. Participants noted that the potential for them

to contribute to speeding up vaccine development was a strong motivation, especially given that they were at risk of contracting the virus in their daily lives. This sentiment is supported by research investigating the attitudes of 1900 volunteers who had registered their interest in participating in a COVID-19 HCT [17]. Volunteers overwhelmingly reported that they were motivated by altruistic goals such as the prospect of helping with research that could speed the recovery from the pandemic.

To our knowledge, only two studies have directly investigated the acceptability of HCTs in large samples [18, 19], and both specifically focused on COVID-19. Broockman and colleagues included mostly US participants and smaller samples from seven other predominantly English-speaking countries. They found high levels of support for the ethical acceptability of HCTs: aggregating across geographic regions, 75% of respondents favored an HCT over a typical vaccine trial. Barker and colleagues used a UK sample, and found high (69%) support for running a COVID-19 HCT in the UK.

Although these studies suggest a high level of support for COVID-19 HCTs among the general public, they have some limitations which can be addressed to produce valuable new insights. The small sample size of Gbesemete and colleagues' focus groups limits the generalizability of the findings. The larger study of Broockman and colleagues may be limited by its framing of the comparison between HCTs and typical trials. HCTs were presented as necessarily reducing the time taken to develop a vaccine and thereby saving hundreds of thousands of lives relative to a typical trial. In reality, such gains are not guaranteed, so the judgments of survey respondents may differ from their perceptions of a real trial in which the benefits are highly uncertain. The UK survey of Barker and colleagues was focused on the UK population and revolved around the specific consideration of a HCT for COVID-19 taking place. All these studies were conducted before the wide availability of vaccines for COVID-19. This may have increased respondents' sense that such trials were urgently needed. Finally, as the studies were focused on the specific use of HCTs in relation to COVID-19, they may be less informative for gauging public perception of HCTs in principle.

In addition to perceptions of the general public, additional insights might be gained by reaching out to a broader range of stakeholders. To the best of our knowledge, no empirical research has examined the attitudes of clinical trials research personnel towards the ethical acceptability of human challenge trials (though Grimwade and colleagues consulted HCT researchers specifically regarding attitudes towards payment for risk). Understanding the attitudes of researchers who are actively involved in clinical trials research can help reveal similarities and differences between the views of the public and professionals, as well as highlight potential barriers–be they ethical or practical–to the wider adoption of HCT methodology among research personnel.

The present study sought to expand upon previous work by assessing public attitudes towards HCTs considered in principle, rather than specifically in relation to COVID-19, among a sample of US adults. We asked about support for various possible uses of HCTs for which support might vary: using them for knowledge generation, for developing vaccines, or developing treatments. We aimed to present HCTs in a more neutral manner than in some previous studies by describing both possible benefits and drawbacks of their use, and explicitly stating that possible benefits were not guaranteed. Furthermore, we sought to understand the importance respondents attributed to several ethically relevant factors, which were identified on the basis of past research and discussion in bioethical literature. Our study was intended to contribute to the growing body of public consultation surrounding HCTs that will be important for conducting HCTs in a manner that will be acceptable to the public, and to understand what aspects of a trial may be most crucial in determining their public acceptability. Finally, we supplemented this consultation of the US public with a small sample of clinical trials

research personnel, to assess views about HCTs among those involved in medical research, and to investigate potential similarities and differences with how the general public views this type of research. This researcher sample was also used to provide insights into the desire to conduct HCTs and possible barriers among researchers to adopting this type of experimental approach.

## Materials and methods

### Ethical approval

All procedures were approved by the Solutions IRB ethical review board (#2022/02/18 & #2023/02/7). The research was performed in accordance with relevant guidelines/regulations. Participants read information about the study and provided informed consent to participate by selecting a multiple choice 'agree' option. Documented consent (i.e., name and signature) was waived by the ethics board on account of increasing the risk of a breach of confidentiality. The enrolment for the public opinion survey took place on November 10th-11th 2022, and for the research personnel between March 21st 2023 and June 27th 2023.

### Recruitment and exclusion criteria

**Public opinion sample.**　Data were collected from US residents aged 18 or above using the *Prolific* platform, and fielded to equal numbers of men and women through quotaing on the platform. Participants received $1.74 for completing the survey. The survey was fielded on November 10th 2022 and completed by 1999 respondents. Participants were excluded if they did not successfully answer all three comprehension checks related to information provided about HCTs (n = 474), or if they identified as non-binary (n = 25: the *American Community Survey*, based upon which we conducted analyses, asks sex as a male/female binary variable, meaning that non-binary respondents cannot be incorporated into these analyses [20]. Sensitivity analyses in which these respondents were included in analyses and assigned as either all male or all female respondents are presented in the S1 and S2 Figs in S1 File, and indicate that the inclusion or exclusion of these respondents would not shift any outcomes). The final sample for analyses was 1500.

**Research personnel.**　To be eligible, research personnel were required to indicate that they were aged 18 or above, fluent in English, and had been involved in conducting Phase III clinical trials for infectious diseases within the last 5 years. Possible research personnel were identified via *ClinicalTrials.gov* and *PubMed*, searching for Phase III clinical trials for a range of infectious diseases taking place between 2017 and 2023 (see S1 File for more detail), as well as direct outreach to prominent vaccine and treatment research centers. This search resulted in 6068 possible email contacts. Outreach was supplemented by a Twitter post from *1DaySooner* and a notification on *Astral Codex Ten*, a blog that was anticipated to include a number of doctors and academic researchers in its readership, and the owner of which funded this research. This outreach included links either directly to the survey or to a landing page giving some information about the study and a link to the survey. Outreach took place between March 21st 2023 and June 27th 2023, and resulted in 385 unique website visits and 67 visitors opening the survey. For the *General support* and *HCT interest and barriers*, we only included those research personnel who reported not having conducted an HCT, resulting in samples of n = 21 and n = 20 for these outcomes respectively, and 20 answering whether HCTs generally should/ should not be allowed. For analyses of *Importance* and *Acceptability* ratings, both those who have and have not conducted HCTs were included, resulting in sample sizes of 33 and 32 respectively.

Information on the research backgrounds of personnel across these different outcomes in presented in the S1 Table in S1 File. In brief, two thirds or more of research personnel reported having PhDs in their field of research, and the majority (88% or more across outcomes) conducted work in an academic/research hospital setting, rather than with industry. The most commonly reported research role was principal investigator, making up approximately 50% of personnel across outcomes. The most frequent countries in which research was based were the USA, UK, and Canada.

**Measures.**   The survey was divided into sections. Both the public and research personnel were presented with the *HCT explanation and comprehension check*, *General support*, *Importance ratings*, and *Acceptability comparisons*. The public then completed *Demographics*. Research personnel were instead shown the *Research Background* and *HCT Interest and Barriers* sections. Full content of each survey is available in the S1 File.

**HCT explanation and comprehension check.**   Participants were provided with an explanation of what an HCT is, highlighting the key feature of purposeful infection with a pathogen. It was noted that all participants take part of their own volition, and that care is taken to select respondents who are not expected to have serious adverse reactions to infection. Respondents were presented with bullet points indicating possible benefits and risks of HCTs (e.g., purposeful exposure could cause harm (risk), there is no guarantee of success (risk), results may not be representative of most vulnerable (risk) vs. lower sample size requirements (benefit), a possible speed up of vaccine or treatment development (benefit)). Whether the benefits or risks were listed first was counterbalanced across participants. On the next page, participants were given three multiple choice questions related to the information they had just read, each with three response options. These comprehension checks had to be answered correctly for participants to be included in analyses.

**General support.**   In three separate questions, participants indicated the extent to which they would support/oppose the use of HCTs for developing new treatments, vaccines, or knowledge about diseases on a 7-point Likert scale ("Strongly oppose", "Oppose", "Slightly oppose", "Neither support nor oppose", "Slightly support", "Support", "Strongly support"). Participants then indicated whether, assuming appropriate precautions were taken, they thought that researchers should be allowed to conduct HCTs ("Yes" vs. "No" vs. "No, and I am against any type of medical research involving humans").

**Importance ratings.**   Participants were requested to indicate how important they thought 15 factors were in deciding whether a particular HCT might be ethically acceptable or not, using a 10-point Likert scale (1 = "Not at all important", 10 = "Extremely important"). The 15 factors, which were presented in randomized order, were: *risk to participants*; *chance participants would catch the illness anyway*; *benefit to wider population*; *effectiveness of existing treatments or vaccines*; *availability of rescue treatments for participants*; *expectation of development speed-up vs. traditional trials*; *possibility of using other study designs to answer the research question*; *participant taking part for money rather than to advance science or help people*; *whether participants just really need the money*; *whether there is payment for risk*; *whether the procedures and results will be made openly available*; *that participants fully understand the risks of taking part*; *whether a placebo control is used* (a definition of a placebo and its purpose were provided); *whether the trial is sponsored or run by a for-profit pharmaceutical company*; *whether participants come disproportionately from disadvantaged groups*.

**Acceptability comparisons.**   To confirm the way in which some factors were important and how much they affect the ethical acceptability of a trial, we asked 5 further questions in which the acceptability of scenarios with or without an ethically relevant feature was varied. Participants indicated whether scenario A or B was more ethically acceptable using a Likert scale format ("A much more acceptable than B", "A more acceptable than B", "A slightly more

acceptable than B", "A and B equally acceptable", "B slightly more acceptable than A", "B more acceptable than A", "B much more acceptable than A"). Scenarios related to the following trial aspects were assessed: *Existence of treatments/vaccines for the disease* (A: There are some treatments or vaccines; B: There are no treatments or vaccines yet); *Payment for risk* (A: Payment is not determined by level of risk, B: Payment is determined by level of risk); *Rescue treatments* (A: There are rescue treatments in case a participant unexpectedly becomes seriously ill, B: There are no rescue treatments); *Placebo control* (A: No placebo control; B: There is a placebo control); *Study sponsorship/independence* (A: The trial will be run by an independent research group, B: The trial will be run by a for-profit pharmaceutical company).

**Demographics.** Participants from the US public answered a range of demographic and attitudinal questions that can be used for conducting multilevel regression and poststratification (MRP) or nationally representative weighting. Note that some items (indicated by asterisks) were included for exploratory purposes and are not included in the pre-registered MRP analyses presented in this paper, and items with a double asterisk are not available in open data: *Sex*, *Age*, *Highest level of education obtained*, *Household income*, *Racial identity*, *US State/ District of residence*, *5 digit ZIP Code\*\**, *Political party identification*, *identification as liberal or conservative\**, *confidence in the scientific community\**, *confidence in medicine\**, *COVID-19 vaccination status\**, *presidential approval*, *general trust in people\**, *views of the Bible\**, *attitudes towards disciplining a child with spanking\**, and *2020 presidential vote\**. Percentages of different demographics in the public sample are presented in the S2 Table in S1 File.

**Research background.** Research personnel were asked in which country their research was primarily based, if they have a PhD in their research field, the context of their research (i.e., academia and/or industry), whether the research covers vaccines, treatments, or prophylactics, and their responsibilities in their research group (e.g., PI, study coordinator, trial physician). Research personnel were also asked if they have ever conducted an HCT, or if they personally know other researchers who have.

**HCT interest and barriers.** Research personnel were asked to indicate their level of interest in conducting an HCT (5-point Likert scale, from "No interest in conducting an HCT" to "A great deal of interest in conducting an HCT"). They were then asked how much interest they would have if there were no logistical barriers to conducting an HCT (e.g., there was funding, approval, staff, and facilities). Research personnel were also asked to what extent several factors were reasons they might not pursue HCT research (on a 5-point Likert scale, from "Not a reason for me at all" to "A very strong reason for me"). Considerations were: *ethical concerns*, *doesn't match career aims/scientific goals*, *difficulty with study approval*, *difficulty with funding*, *difficulty obtaining facilities*, *concerns over ramifications if something went wrong*, and *seeing no benefit of HCTs vs. other approaches*. Research personnel were given the opportunity to provide additional comments and make suggestions for other researchers we could contact at the end of the survey.

**Pre-registration and data availability.** For general public opinion, analyses and exclusion criteria were pre-registered on the *Open Science Framework* (https://doi.org/10.17605/OSF.IO/ WVQ9N). For regression models of ordinal items, we made a minor modification of the pre-registered approach, widening the prior on the intercepts, from normal(0, 2) to normal(0, 4). This was because some extreme ordinal ratings (e.g., almost no one assigning low importance to items about participant safety) seemed to cause problems with MCMC sampling. This change does not apply to comparisons between items, nor favor higher vs. lower ratings.

The analytic approach for the sample of research personnel was not pre-registered, but the form of all comparisons made between items is the same as that for the public sample, only using a form of randomization testing, rather than MRP.

Both the general public sample data and research personnel data are available at https://osf.io/u2ev9/?view_only=c971b7aa71ed4675ae34856797d31bfd.

## Analyses for public opinion

**Multilevel regression and poststratification.** We used Bayesian MRP to generate nationally representative estimates for the US adult population based upon our sample data. The sampling for the Bayesian regression models used Hamiltonian Monte Carlo (HMC) No-U-Turn sampling (NUTS). MRP [21] is an approach that can be used to produce accurate population-level estimates from samples that are not strictly representative or even highly unrepresentative [22–24]. The two step MRP procedure involves: 1) conducting a multilevel regression, in which the outcome of interest is predicted in a multilevel regression model with a range of demographic features as predictors, and 2) a poststratification step, in which the regression model is used to predict the responses of the target population based upon the number of people with different combinations of those demographic features in the population of interest.

To estimate the US population-level opinion, we included: Age (18–24, 25–34, 35–44, 45–64, 65+), Sex (Male, Female), Education (High school graduate/equivalent or less; Some college, no degree; Graduated from college; Completed graduate degree), Household income (<\$20,000, \$20,000-\$49,999, \$50,000-\$79,999, \$80,000-\$99,000, \$100,000-\$150,000, >\$150,000), Race (Asian/Asian American; Black/African American; Hispanic/Latino; White/Caucasian; Other), political party identification (Republican, Democrat, Independent–Independent included those indicating they were unsure or unaffiliated), State/District of Residence, US Census region of that State, and *z*-scored State-level vaccine hesitancy rate as of November 2$^{nd}$ 2022 [25]. For poststratification, we used the 5-year 2020 American Community Survey [26] to generate a cross-tabulation of all these features in the US adult population. This poststratification table was then extended to include party identification using MRP on data from the 2020 Cooperative Election Study [27] to estimate political party identification depending on demographics.

The following regression formula was used for all analyses, where (1 | *variable*) indicates that each level of that variable was informed by and modeled as a deflection from the average of all the levels:

Outcome ~ Sex + (1 | Age) + (1 | Education) + (1 | Income) + (1 | Race) + (1 | Party) + (1 | State) + (1 | Sex*Race) + (1 | Education*Age) + Region + State_Vaccine_Hesitancy

That our MRP approach can capture and account for possible selection biases in the respondents is supported by MRP modeling of the *Presidential Approval* variable. In the raw data, approval was 53.1%, and disapproval 40.1%. On the date the survey was conducted, polling aggregator *FiveThirtyEight* estimated approval to be at 41.2%, and disapproval to be at 53.5% [28]. Following MRP, the population estimate of approval based upon our data was 40.4% [37.3–43.7] approval, and 52.7% [49.2–56.0] disapproval, well within error margins suggested by *FiveThirtyEight*.

Likert scale responses were analyzed using cumulative ordinal regression with a probit link. Categorical outcomes were analyzed using categorical regression with a logit link. Comparisons among Likert responses were analyzed through assessments of within subjects *Probability of Superiority* (*PSup*) [29]. *PSup* is a probability-based measure of effect size ranging from 0 to 1, indicating the proportion of respondents for whom one would expect their rating for one item to be higher than the other: .5 indicates equivalence of the items, with values approaching 1 indicating increasing probabilities that item A is rated higher than item B, and values approaching 0 indicating increasing probabilities that A is rated lower than item B. This measure is appropriate for the ordinal items used here, as it does not assume the Likert ratings

reflect a scale response. For each pairwise item comparison under consideration, a 3-level categorical variable was generated indicating item A of the pair being greater than, equal to, or less than item B of the pair in each respondent. MRP was then conducted on this categorical variable to estimate population level *PSup* for that comparison.

Weakly informative priors used in analyses are detailed in the S1 File.

**Ranges of practical equivalence.**   For determining whether comparisons among items reflected a reliable and non-negligible difference in ratings, a range of practical equivalence (ROPE) was set from .47 to .53 for *PSup* comparisons, outside of which the 95% highest density interval (HDI) of the posterior estimate would have to fall in order to be considered reliably different [30]. Additional effect size thresholds can be considered at .6 (a 'supermajority' superiority for one item over the other), .67 (a 2:1 ratio of superiority), and .75 (a 3:1 ratio of superiority). A simulation-based assessment presented in the S1 File indicated that a ROPE of .47-.53 should greatly constrain the possibility of false positives (concluding that there is a difference between paired ratings when there is none) across comparisons to less than 1% of comparisons made, if binary judgments of presence vs. absence of effects had to be made on the basis of this ROPE (S3 and S4 Figs in S1 File). Note that we favor considering the relative magnitudes of differences and the uncertainty around them rather than making binary classifications.

**Analyses for research personnel.**   Given the relatively small sample of research personnel and low response rate with possible selection bias, it is not appropriate to assume that our sample of research personnel can be analyzed as a random sample from the clinical trials personnel population. Hence, we primarily present results from the sample descriptively, without confidence intervals derived from random sampling assumptions.

**Randomization tests.**   Statistical tests can still be performed using randomization testing to determine the likely presence of significant differences in within subjects ratings among the research personnel [31]. For any pairwise comparison of the form A vs. B, we calculate the observed *PSup* A > B in the sample. We then perform 1000 permutations in which paired scores from A and B are randomly switched, and calculate the percentage of times we achieve a *PSup* as or more extreme than that observed (aka. The *chance* or $c$ value). For each set of pairwise comparisons (e.g., comparing all the importance ratings with one another), we can select those comparisons with the lowest $c$ values up to a cumulative chance level of 5%, aiming to hold the false discovery rate across that set of comparisons to 5%. A simulation-based assessment presented in the S1 File indicated that this approach should constrain the expected percentage of false positives for each set of comparisons (e.g., among *Importance Ratings*, or among *HCT Barriers*) to below 5% (S5 Fig in S1 File).

**Comparisons between research personnel and US population estimates.**   To compare research personnel with the general population, we use the posterior distribution of expected responses for the respective item from the general public and compute the *PSup* value for these estimated responses being superior to the raw responses from the research personnel. These comparisons thus take into account uncertainty in the public opinion estimates, but treat the research personnel sample as fixed. Hence, the comparisons are specifically for this sample of personnel and do not represent differences between the general population and *all* infectious disease clinical trials personnel.

## Results

### Support vs. opposition for HCTs

Both the general population and our sample of research personnel were mostly supportive of the use of HCTs for developing treatments, vaccines, and knowledge about infectious diseases

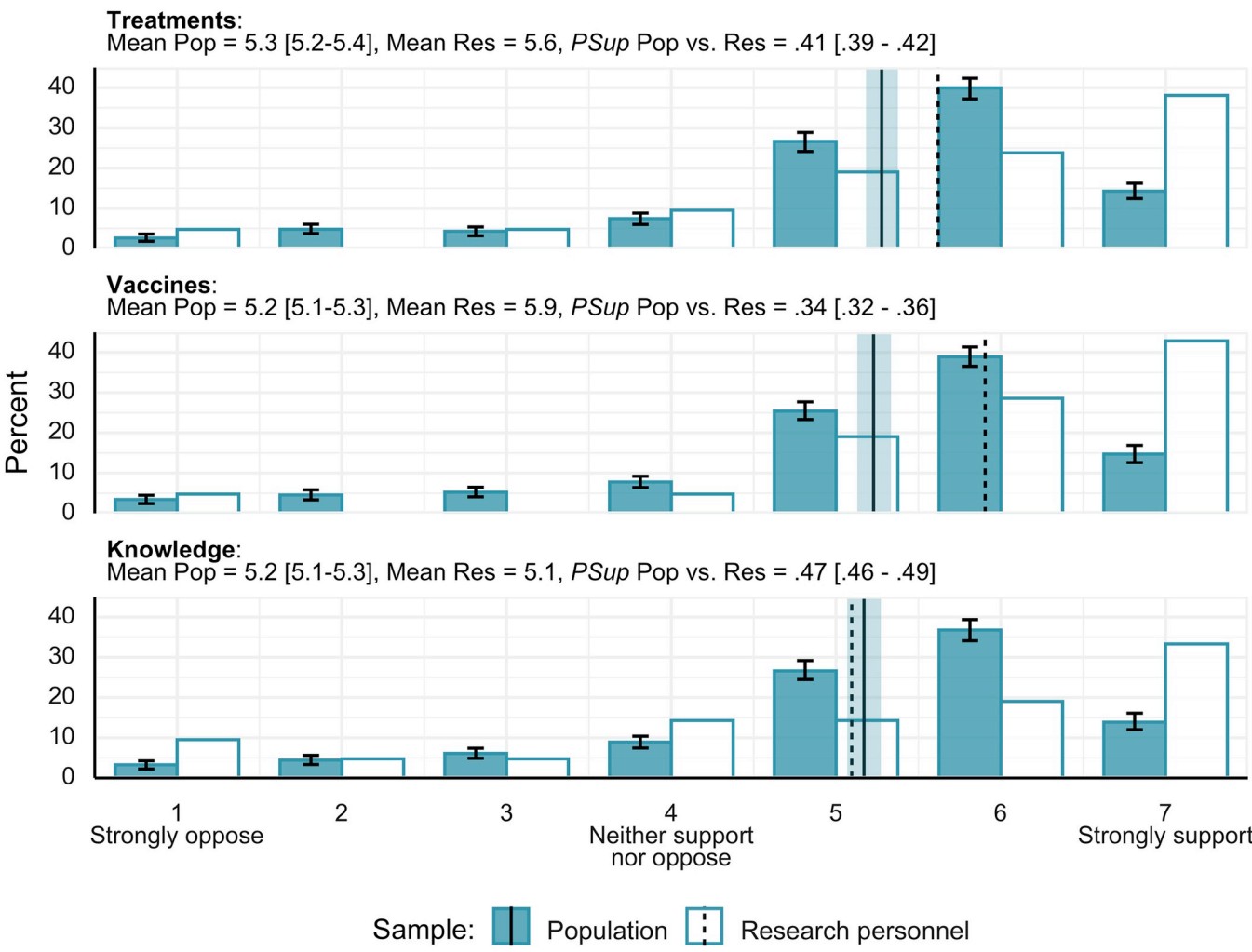

**Fig 1. Support vs. opposition for different types of human challenge trial.** Both the general public and our research personnel sample were largely supportive of using HCTs for treatments, vaccines, and knowledge development. Pop = general population, Res = research personnel; *PSup* = probability of superiority. Error bars and shaded region around mean line represent 95% highest density interval. Vertical lines represent means of the respective ratings.

(Fig 1). Reducing the responses down to a simple support vs. neutral vs. oppose, we estimate: For treatments, support of 80.9% [78.5% - 83.2%], neutral 7.4% [6.0% - 8.8%], and opposition 11.7% [9.9% - 13.8%]; for vaccines, support of 79.0% [76.6% - 81.6%], neutral 7.8% [6.4% - 9.2%], and opposition 13.2% [11.0% - 15.1%]; and for knowledge, support of 77.3% [74.4% - 79.9%], neutral 8.9% [7.4% - 10.4%], and opposition 13.8% [11.7% - 16.0%]. Among the research personnel the percentage support was also high relative to opposition (Treatments = 81.0% vs. 9.5% vs. 9.5%; Vaccines = 90.5% vs. 4.8% vs. 4.8%; Knowledge = 66.7% vs. 14.3% vs. 19.0%—with ordering reflecting Support vs. Neutral vs. Opposition).

*PSup* estimates suggested comparable levels of support among our research personnel and the general population for using HCTs to develop new knowledge about diseases, whereas there was a tendency for the researchers to be more supportive than the general population regarding HCTs in developing vaccines and treatments. For comparisons of support for the different uses of HCTs *within* the general population, we did not find any robust differences: all comparisons showed *PSup* values crossing or fully within our prespecified ROPE boundaries for the public (*PSup*$_{Within}$ for the general population: Treatments—Vaccines = .51 [.50 -

.53], Treatments—Knowledge = .54 [.52 - .56], Vaccines–Knowledge = .53 [.51 - .54]). The research personnel showed a marginally significant preference for the use of HCTs to develop vaccines, relative to knowledge generation (Treatments—Vaccines = .40, $c$ = 27.6%, Treatments–Knowledge = .60, $c$ = 35.0%, Vaccines–Knowledge = .69, $c$ = 4.25%), though it should be noted this difference was not expected *a priori* and is close to the 5% significance threshold.

Estimates for the general public (Support = 86.4% [84.1% - 88.9%]; Opposition = 10.1% [8.4% - 12.5%]) and for research personnel (Support = 90.0%; Opposition = 10.0%) suggested a large majority supported the general idea that researchers should be allowed to conduct HCTs, assuming appropriate ethical oversight. An estimated 3.4% [2.2% - 4.7%] of the general population would be opposed to any kind of medical research involving humans, indicating an opposition that is not specific to HCTs.

## Importance of factors in determining ethical acceptability

**Importance ratings.** For both the general population and research personnel, the most important factors were the risks faced by participants, and that participants understood these risks (Fig 2, for Likert-style response plots, see S6 Fig in S1 File). A plot showing the proportion of respondents giving each response option for each question is presented in the Supplementary Materials. The general population was estimated to have the least concern over participant motivations–whether they "just really need the money" or are doing the study for money

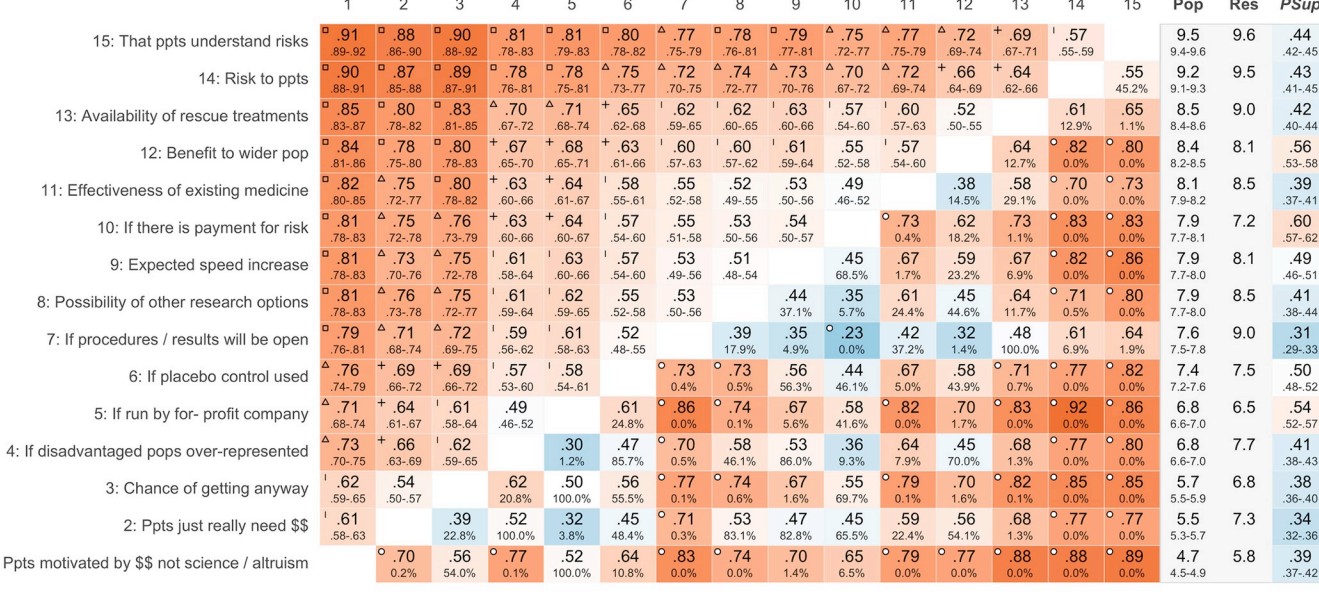

**Probability of Superiority (PSup)**
Above diagonal: US public - probability that Y axis item (items on left) > X axis item (items on top)
Below diagonal: Research personnel - probability that X axis item (items on top) > Y axis item (items on left)
Right-hand PSup column: Probability that US public rating > Research personnel sample rating

**Fig 2. Matrix of within subjects probability of superiority (*PSup*) for the importance of different factors in determining ethical acceptability of an HCT.** Above diagonal is general population estimates, below diagonal is the researcher sample. Bold columns on the right show mean importance ratings and *PSup* for general population (Pop) vs. research personnel (Res). For research personnel, percentages indicate chance of getting a result as or more extreme with a permutation test, and circles indicate within subjects differences that fall within a cumulative 5% error rate. Intervals represent 95% highest density intervals, and shapes show increasing effect sizes excluded from the interval: | = .53 (non-negligible small difference), + = .6 (supermajority preference), Δ = .67 (2:1 ratio), □ = .75 (3:1 ratio).

rather than altruism or to benefit science. For most items, research personnel tended to give higher importance ratings than the general population. Relative to the general population, our research personnel were particularly more concerned that procedures and results would be openly available, which research personnel also considered to be more important than payment for risk.

**Acceptability comparisons.** Regarding the acceptability of HCTs conditional on the presence/absence of effective treatments/vaccines and of rescue treatments, ratings from the general public and from the research personnel were very similar (Fig 3). Moreover, both the

**Rescue treatment availability** Mean Pop = 5.8 [5.7-6.0], Mean Res = 5.9
A = Effective treatments for the disease already exist, and so can be given to a participant if they unexpectedly become seriously ill
B = Effect treatments for the disease do not exist, and so cannot be given to a participant if they unexpectedly become seriously ill

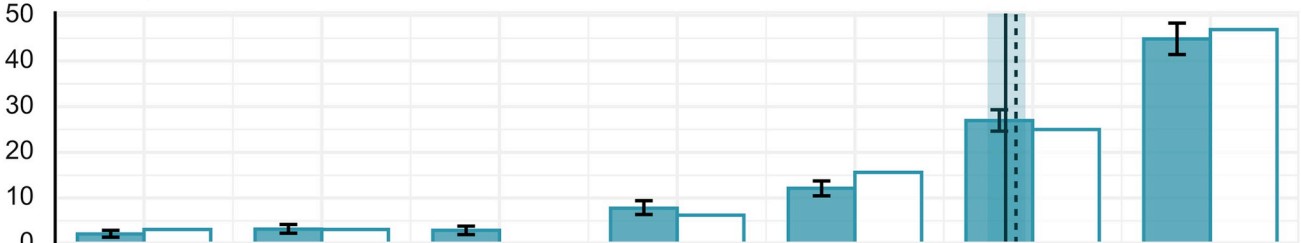

**Existing treatments/vaccines** Mean Pop = 4.3 [4.2-4.5], Mean Res = 4.4
A = There are already some treatments or vaccines for the disease
B = There are no treatments or vaccines for the disease yet

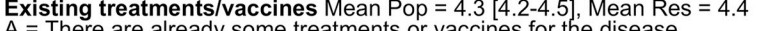
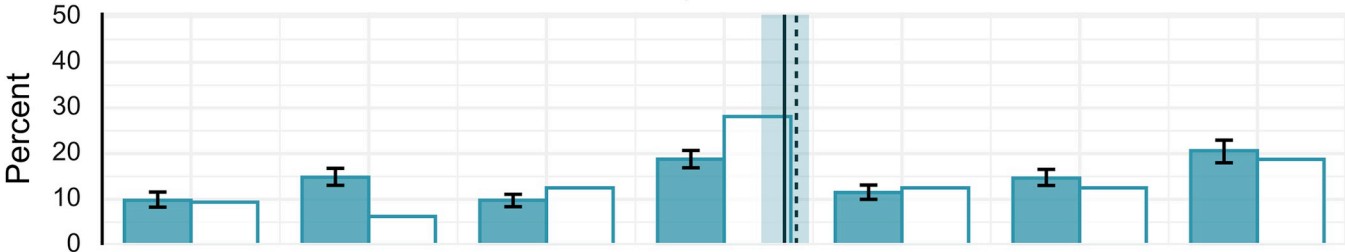

**Payment for risk** Mean Pop = 2.2 [2.1-2.3], Mean Res = 3.5
A = Payment for participants is not determined by the level of risk involved in taking part
B = Payment for participants is determined by the level of risk involved in taking part (so higher risk means more payment)

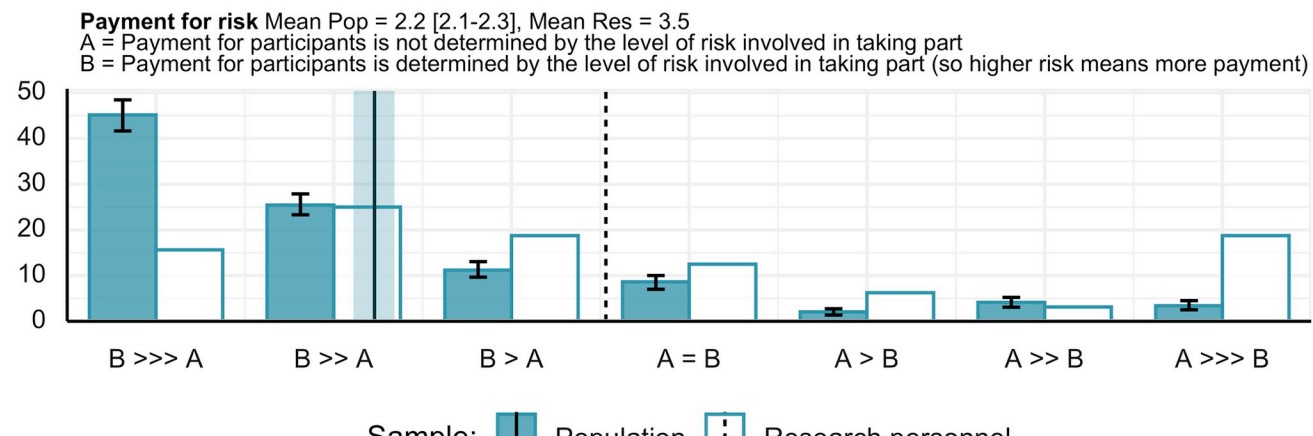

**Fig 3. Relative ethical acceptability for different HCT scenarios.** >>> much more acceptable, >> more acceptable, > slightly more acceptable, = equally acceptable. Error bars and shaded region around mean line represent 95% highest density interval. Pop = general population, Res = research personnel. On numeric scale for means, ratings go from 1 to 7. Vertical lines represent means of the respective ratings.

general population and research personnel appeared similarly susceptible to a framing effect that was hypothesized in our preregistered analysis plan. Specifically, respondents were roughly neutral with respect to whether it is more or less acceptable to conduct an HCT in the presence vs. absence of existing treatments when this was referred to generally as just there being vs. not being existing treatments and vaccines available. In contrast, when the presence vs. absence of treatments is framed more specifically as there being vs. not being *rescue* treatments, ratings indicated that the scenario in which treatments were available was clearly more ethically acceptable. In the general population, *PSup* for this framing difference was .75 [.73 - .78], which is very similar to the *PSup* of .73 (*c* level = 0.1%) among research personnel.

Regarding payment for risk, the general public showed a substantially larger preference for when there is vs. is not payment for risk than did the research personnel, with a public vs. research personnel *PSup* of .30 [.28 - .32], where the inferiority indicates greater preference for payment for risk among the general public than among the research personnel.

Two additional items for acceptability, whether a placebo control is used, and whether the trial is run by a for profit pharmaceutical company vs. an independent research group, are presented in the S7 Fig in S1 File. Both the general public and research personnel were predominantly either neutral, or tended to favor both the use of a placebo control (vs. no placebo control) and a trial being run by an independent research group (vs. by a for-profit pharmaceutical company).

**Interest in HCTs and barriers to involvement.** Among researchers who had not already been involved in an HCT, interest in conducting one was relatively low (Fig 4). However, researchers gave a higher interest rating when asked what their interest would be if there were no logistical barriers to conducting an HCT. When asked to rate various possible reasons for not conducting an HCT, very few researchers indicated that they did not think there were any

### Interest in conducting HCT
Mean current = 2.8, Mean with no barriers = 3.4, *PSup* = .67 *c* = .0%

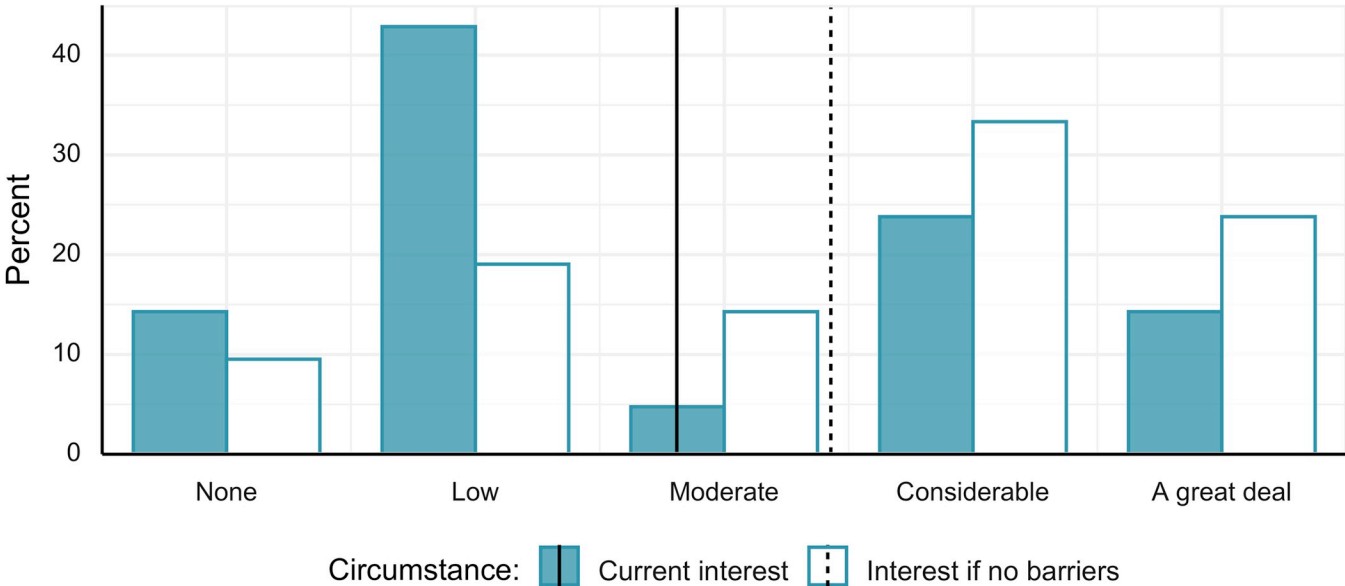

**Fig 4. Interest in conducting an HCT was higher when researchers were asked to imagine a situation in which there were no logistical barriers to conducting an HCT.** On numeric scale for means, ratings go from 1 to 5. Vertical lines represent means of the respective ratings if treating the ordinal items as numeric.

benefits to conducting an HCT: in pairwise *PSup* comparisons among items (presented in the S8 and S9 Figs in S1 File), this item was reliably rated lower than all others. The most-endorsed barriers were primarily logistical, most notably expecting difficulties with ethical/institutional review boards.

## Discussion

We found high levels of public support for the use of HCTs, whether considering their use for vaccine development, treatment development, or generating new knowledge in relation to infectious diseases. Across these different types of trials, support ranged from 77% to 81%, with opposition ranging from 12%-14%. These high levels accord with a previous study of public attitudes towards HCTs, which found that approximately three quarters of respondents preferred an HCT over a typical vaccine trial for COVID-19 vaccine development [19], and a survey of the UK public which found 70% support for COVID-19 HCTs [18]. As noted, the study of Broockman and colleagues' may have been slightly biased by the suggestion of large and definite benefits of an HCT approach. In the present study, we expressly conveyed that there were no guarantees that an HCT would successfully produce a treatment or vaccine. That we still found high levels of support for HCTs suggests that HCTs may have general acceptability, at least when the reasoning and potential advantages of the approach are explained to respondents. Our sample of research personnel was also supportive of HCTs, with support ranging from 67% (for knowledge development) to 90% (for developing vaccines).

Beyond assessing general support for the use of HCTs, we expanded upon previous research by assessing the importance attributed to a range of factors that may make an HCT more or less acceptable. Foremost among the items in importance for both the general public and research personnel were two factors that directly related to the risks posed to participants: the level of risk exposure and that participants fully understand those risks, and a third item that may be seen to relate to the risks participants face: the availability of rescue treatments. Among the general public, levels of risk, and that participants understand said risk, were estimated to be of reliably greater importance than concerns regarding the 'greater good', such as the potential benefit to wider society or expected increase in the speed of treatment or vaccine development. The least important considerations among the general public were both related to participants' motivations for taking part, with concern over whether participants are motivated by money rather than by altruistic motivations being rated as lower than any other concern. Research personnel also rated participants being motivated by money rather than altruistic or other laudable concerns to be of least importance among the different factors, but showed greater concern than the general public about participants only taking part because they "just really need the money". Our research personnel sample also seemed particularly more concerned than the general population about procedures and results being open.

Although we estimated that the public granted relatively low importance to whether participants were motivated by monetary incentives, they deemed it important whether or not payment scaled with the risks participants faced, and rated this as of higher importance than did the research personnel. This accords with the findings of Grimwade and colleagues [34], who found both public support for payment for risk, and that it was more supported among the general public than among HCT researchers. Our sample of research personnel tended to mostly agree that a scenario in which payment scaled with risk would be preferable to one in which it did not, but approximately 1 in 5 reported very strongly preferring a situation in which payment for risk did not occur. Positive public and even researcher attitudes towards payment for risk do not, of course, mean that exorbitant payments should be used or that

concerns over undue inducement are completely unwarranted. Discussion among ethicists has suggested that payment for risk could be unethical due to increasing the risk of exploitation, or of participants withholding important risk-related information in order to be included in a high-paying study [32]. However, large discrepancies between public attitudes and what may be seen as the prevailing wisdom or norm espoused in bioethical discussion and existing payment practices may highlight a need for better justification of the case against payment for risk, or else of changing the status quo [33, 34]. Indeed, among a sample of potential HCT volunteers, greater concern was expressed about exploitation through too little rather than too much pay [33].

In addition to payment for risk, we were able to evaluate some considerations that we believe to be previously unexplored. Firstly, given our expectation that risk to participants would score highly as a concern, we wondered whether respondents would be averse to the use of a placebo control, as this would mean that participants in an HCT would at least temporarily remain untreated or unprotected with the potential treatment or vaccine. In comparing a trial with or without a placebo control group, we estimate that the general public would find a trial with a placebo control group to be more acceptable than one without, with another spike in responses indicating no preference with respect to there being vs. not being a placebo control group. The pattern of responses was similar among our research personnel sample.

Next, considering the potential for mistrust of pharmaceutical companies [35], we assessed respondents' possible concerns regarding study sponsorship and management. In terms of importance, whether or not the trial was run by an independent research group ranked around the bottom third of concerns, though still received an average rating of 6.8/10 in the public, and 6.5/10 among research personnel. In comparing the ethical acceptability of a trial run by an independent research group vs. a for-profit pharmaceutical company, we estimated that the vast majority of the US public would either have no preference (~25%) or prefer that the trial was run by an independent research group (~70%). In terms of public perception then, it would seem that challenge study coordinators should consider carefully their partnerships with industry, owing to the potential for undermining public trust. Our sample of research personnel similarly indicated either no preference or a preference in favor of an independent research group.

One final comparison of note was in respondents' judgments of the relative acceptability of two different framings of situations related to the absence of existing treatments when a trial takes place. In the first framing, respondents rated the acceptability of an HCT when there were no existing treatments or vaccines vs. when there were some (henceforth the *general* framing). In the second framing, respondents rated the acceptability of an HCT when there were vs. were not *rescue* treatments available should a participant unexpectedly become seriously ill (henceforth the *rescue* framing). These do not represent an absolutely clean comparison of framings (e.g., one mentions specifically there being no rescue treatments, whereas the other mentions there generally being no existing treatments or vaccines, with the mentioning of vaccines possibly also affecting responses), but we believe the comparison between the two remains instructive. Respondents showed a quite even spread of preferences to the general framing, whereas for the rescue framing, respondents overwhelmingly found it more acceptable in the case where there *were* rescue treatments. This was the case amongst both the general public and research personnel. This difference in preferences was large, with about three quarters of people estimated to find the absence of treatments less acceptable when framed as an absence of rescue treatments specifically vs. a more general statement about there being no current effective treatments or vaccines. We suggest that when people are thinking of there being no existing treatments or vaccines generally available, they might consider how this could make an HCT even more beneficial in terms of the general good, as it could produce a

novel intervention with significant benefits. However, respondents may not readily recognize the possibility that if there really were *no* existing effective treatments, then participants in the trial could end up with an incurable illness. When this possibility is made explicit, respondents may focus more on the highly identifiable risk to each participant in a trial, and thus find this situation much more concerning. This fits with the very high importance rating given to the objective risk faced by participants in a trial. It might be beneficial to make stakeholders more explicitly aware of such concerns, where they apply, to ensure that people do not feel they were misled if messaging about HCTs focused only on the upsides of running such trials when there are no existing treatments.

Our sample of research personnel answered several additional questions regarding their desire to utilize HCT methodology in their own lines of research. Under current circumstances, the modal response was of low interest in conducting an HCT. However, it is possible that facilitating HCT research via removing real or perceived barriers to their implementation could increase the number of researchers who would be interested in using HCTs, as interest was increased conditional on there being less logistical barriers. Although research personnel appeared to see possible benefits of HCTs, perceived difficulties in getting ethical approval, funding, and facilities necessary for HCTs were rated as quite strong barriers to their utilization.

## Limitations

The perceived acceptability of different measures to combat infectious diseases may vary with the course of current events, and it is possible that our findings reflect a specific set of perspectives arising in the aftermath of a recent pandemic. We do not think this detracts from the value of taking such a snapshot of opinion, but simply highlight that there could be shifts in perspectives in the future. Furthermore, our findings represent support of HCTs *as presented to them in the format we provided*. We aimed for this presentation to convey the reasoning behind the use of HCTs, and to describe both risks and benefits associated with their use in a neutral manner. When we consider population level support of HCTs 'in the wild', we must contend with the possibility–perhaps even the likelihood–that subpopulations will receive information about HCTs in formats that are more contentious or biased than that which we used. For example, if there were a campaign against HCTs that involved highly negative messaging and only conveyed drawbacks, then we might expect some people to be correspondingly swayed against the use of HCTs. Similarly, messaging suggesting a disease is especially dangerous, or conversely that it is benign or even a hoax, might also affect how people view the value of any particular HCT. Hence, the estimates provided may be somewhat optimistic, in that they reflect the opinions of respondents who likely have relatively few if any preconceived notions regarding HCTs, and who were willing to read the information provided. A potential implication of these considerations is that those wishing to retain the high level of support for HCTs in the population that seems possible based on our data might ideally engage and get on-board stakeholders from across the political and ideological spectrum, to reduce the chances that views of HCTs become subject to polarizing campaigns.

We would also stress that neither public nor professional support of HCTs is proof of their ethical acceptability, which is a question of ethical principles. However, pervasive public *opposition* to HCTs could have been an argument against their ethical acceptability, given that public trust and acceptance of the procedures used to generate new treatments and vaccines may be necessary for their uptake. If the public will not use that which has come from HCTs owing to mistrust and disapproval then the risks taken by HCT participants may not be balanced by public benefits. Several authors on the topic have therefore highlighted the importance of public consultation in relation to HCTs, of which this research forms a part [4].

Finally, as with all attempts to represent the opinions of different populations with samples from those populations, there is always a possibility of unrepresentative sampling. Our assessment of our MRP procedure via a measure of presidential approval suggests that we were able to correct for obtaining a skewed sample from the public, but there may be demographic features that we are not able to correct for but which do relate to attitudes towards HCTs. With respect to our sample of research personnel, we suggest that the sample should not be taken as representative of all personnel involved in phase III trials for infectious diseases, and as such the generalizability of answers from the research personnel is particularly open to question. Nevertheless, comparing this specific sample with the general population estimates can give some indications as to possible ways in which researchers more broadly might diverge from or align with general population attitudes: for example, it makes sense that research personnel might be especially concerned with the openness of scientific procedures or, given the current ethical status quo in academic research, less positive about payment for risk than the general population.

## Conclusion

In summary, we estimate that US adults are supportive of the use of HCTs in aiding the development of new treatments, vaccines, and generating new knowledge–as were a sample of research personnel. The foremost concerns were risks faced by participants, and their full comprehension of these risks. Our results accord with and build upon previous studies suggesting broad public support for HCTs specifically in relation to COVID-19 [18, 19], showing that such support may be more general. When conducting HCTs or engaging in public discourse, we suggest that researchers can consider the importance the public attributes to the factors we assessed. Though some such factors may seem obvious (people care greatly about participant risk), some perceptions may highlight the possibility of revisiting some ethical perspectives currently considered the status quo [34]: US adults tended to believe that participants should be paid in accordance with the risks they are exposed to, and are relatively unconcerned about their motivations for participation.

## Supporting information

**S1 File. Supporting information.** File containing all supplementary figures and additional information on the analytic approach and raw questionnaire.
(DOCX)

## Author Contributions

**Conceptualization:** James William Benjamin Elsey, David Manheim, Abigail Marsh, Virginia Schmit, David Moss.

**Data curation:** James William Benjamin Elsey, David Moss.

**Formal analysis:** James William Benjamin Elsey, David Moss.

**Funding acquisition:** David Manheim, Virginia Schmit.

**Investigation:** James William Benjamin Elsey, David Moss.

**Methodology:** James William Benjamin Elsey, David Moss.

**Project administration:** David Moss.

**Supervision:** David Moss.

**Visualization:** James William Benjamin Elsey.

**Writing – original draft:** James William Benjamin Elsey, David Moss.

**Writing – review & editing:** James William Benjamin Elsey, David Manheim, Abigail Marsh, Virginia Schmit, David Moss.

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
