## [Decision Letter · Decision Letter 0]

14 May 2024

PONE-D-23-31643Ethical Acceptability of Human Challenge Trials: Consultation with the US public and with research personnelPLOS ONE

Dear Dr. Elsey,

Thank you for submitting your manuscript to PLOS ONE. After careful consideration, we feel that it has merit but does not fully meet PLOS ONE’s publication criteria as it currently stands. Therefore, we invite you to submit a revised version of the manuscript that addresses the points raised during the review process.

We look forward to receiving your revised manuscript.

Kind regards,

Isha Amatya, M.D.

Guest Editor

PLOS ONE

 [Astral Codex 10 Grant].  

[This work was supported by an ACX Grant to 1Day Sooner, a 501(c)(3) non-profit organization that advocates for people who participate and want to participate in high-impact medical studies, including HCTs. Rethink Priorities was contracted by 1Day Sooner to execute this research project and collected and analyzed data independently of 1Day Sooner.]. 

5. Please remove your figures from within your manuscript file, leaving only the individual TIFF/EPS image files, uploaded separately. These will be automatically included in the reviewers’ PDF.

Reviewers' comments:

Reviewer's Responses to Questions

**Comments to the Author**

1. Is the manuscript technically sound, and do the data support the conclusions?

Reviewer #1: Yes

Reviewer #2: Yes

2. Has the statistical analysis been performed appropriately and rigorously? 

Reviewer #1: Yes

Reviewer #2: Yes

3. Have the authors made all data underlying the findings in their manuscript fully available?

Reviewer #1: Yes

Reviewer #2: Yes

4. Is the manuscript presented in an intelligible fashion and written in standard English?

Reviewer #1: Yes

Reviewer #2: Yes

5. Review Comments to the Author

Reviewer #1: The paper overall is well-presented. Many doubts were explained in the limitations. However, I would recommend that the authors include a socio-demographic table summarizing the characteristics of the public participants surveyed, either in the main manuscript or as extended data.

Reviewer #2: The manuscript is technically sound and the data supports the conclusion along with appropriate statistical analysis. However, I have few points for minor review of the manuscript. They are mentioned below.

1) Please mention how the general participants were fielded in male and female, what was the sampling method used and how they were equally distributed? [article line number 148]

2) It has been mentioned that non binary population were excluded but sensitivity analysis was done to check any major differences by assigning them as male/ female. What method was used to assign them as male/female. And were they also equally assigned as male and female (in numbers)? what method was used for equal distribution? [Article line number 154]

3) The 3rd inclusion criteria for research personnel as mentioned in the manuscript and information sheet in the supplementary data does not match with the research personnel actually involved in the study. Please specify the reason for including participants who do not fall under the mentioned inclusion criteria. [ Article line number 159,160 and 171]

4) The study shows two different groups. It is advisable to mention them by their group name rather than respondents as it would sound less confusing for readers.

5) Please mention the full form of MCMC sampling in the text. [Article line number 272]

6) The total number of participants (N) should be mentioned in the table number 1 provided in the supplementary data with heading "sample information for research personnel across different outcomes" for different outcomes category.

6. PLOS authors have the option to publish the peer review history of their article (what does this mean?). If published, this will include your full peer review and any attached files.

Reviewer #1: **Yes: **Sushila Paudel

Reviewer #2: **Yes: **Dr. Shruti Shah

---

## [Author Response · Author response to Decision Letter 0]

2 Jul 2024

Dear Editor, Dr Shruti Shah, and Sushila Paudel

Thank you for your positive review of our work. We appreciate the time and effort you took to evaluate our manuscript. In the following pages, we have provided responses to any outstanding questions and comments you had, highlighting where any edits have been made in the manuscript.

Editor notes/comments:

The manuscript has now been updated to match the style requirements of PLOS ONE.

Data and analysis files are available via the Open Science Framework (https://osf.io/u2ev9/) and is referred to in the manuscript.

 [Astral Codex 10 Grant]. Please state what role the funders took in the study. If the funders had no role, please state: "The funders had no role in study design, data collection and analysis, decision to publish, or preparation of the manuscript." If this statement is not correct you must amend it as needed. Please include this amended Role of Funder statement in your cover letter; we will change the online submission form on your behalf.

We’ve now updated our funding statement at the end of the manuscript and cover letter:

“ACX had no role in study design, analysis, decision to publish, or preparation of the manuscript. As noted in the manuscript, ACX played a minor role in data collection by increasing awareness of the study among potential research personnel via a mention in a blog post.”

[This work was supported by an ACX Grant to 1Day Sooner, a 501(c)(3) non-profit organization that advocates for people who participate and want to participate in high-impact medical studies, including HCTs. Rethink Priorities was contracted by 1Day Sooner to execute this research project and collected and analyzed data independently of 1Day Sooner.]. 

Please confirm that this does not alter your adherence to all PLOS ONE policies on sharing data and materials, by including the following statement: "This does not alter our adherence to PLOS ONE policies on sharing data and materials.” 

We’ve now added this to our COI statement. In full, the COI is now:

“This work was supported by an ACX Grant to 1Day Sooner, a 501(c)(3) non-profit organization that advocates for people who participate and want to participate in high-impact medical studies, including HCTs. Rethink Priorities was contracted by 1Day Sooner to execute this research project and collected and analyzed data independently of 1Day Sooner. This does not alter our adherence to PLOS ONE policies on sharing data and materials. ACX had no role in study design, analysis, decision to publish, or preparation of the manuscript. As noted in the manuscript, ACX played a minor role in data collection by increasing awareness of the study among potential research personnel via a mention in a blog post.”

4. PLOS requires an ORCID iD for the corresponding author in Editorial Manager on papers submitted after December 6th, 2016. 

I’ve now generated an ORCID id and linked it to the submission.

5. Please remove your figures from within your manuscript file, leaving only the individual TIFF/EPS image files, uploaded separately. These will be automatically included in the reviewers’ PDF.

We’ve removed the Figures from the file itself, and now generated and uploaded TIFF files (including putting them through the PACE system).

6. Please review your reference list to ensure that it is complete and correct

The reference list has been updated and checked for including all the appropriate articles.

Reviewers' comments:

Reviewer #1: The paper overall is well-presented. Many doubts were explained in the limitations. However, I would recommend that the authors include a socio-demographic table summarizing the characteristics of the public participants surveyed, either in the main manuscript or as extended data.

Thank you for this suggestion, including the raw demographics can add to the transparency of information regarding the sample. We’ve now included a demographics table titled ‘Sample information for general population’ in our appendix/supplementary materials. We also highlight that this table is available in the demographics section of the main manuscript.

Reviewer #2: The manuscript is technically sound and the data supports the conclusion along with appropriate statistical analysis. However, I have few points for minor review of the manuscript. They are mentioned below.

1) Please mention how the general participants were fielded in male and female, what was the sampling method used and how they were equally distributed? [article line number 148]

We’ve now slightly modified the sentence here – the participants were recruited via the survey platform called Prolific. In order to ensure an equal number of male and female respondents, you can set a quota to specifically ask for certain percentages on some demographic features, and so we set it to field to 50-50 male and female respondents. On p7, we now say:

“Data were collected from US residents aged 18 or above using the Prolific platform, and fielded to equal numbers of men and women through quotaing on the platform.”

2) It has been mentioned that non binary population were excluded but sensitivity analysis was done to check any major differences by assigning them as male/ female. What method was used to assign them as male/female. And were they also equally assigned as male and female (in numbers)? what method was used for equal distribution? [Article line number 154]

In order to do this, we avoided splitting the respondents between male or female identifications and just ran different regression models either with them all assigned to male, or all assigned to female. Essentially what we found is that this small amount of respondents did not perceptibly or significantly change any of the outcome metrics. The same would be the case if we had tried splitting them in some way across male or female, as then the sample changes would be even smaller. We’ve tried to clarify the nature of the sensitivity analyses with some minor edits on page 8:

“Sensitivity analyses in which these respondents were included in analyses and assigned as either all male or all female respondents are presented in the Supplementary Materials, and indicate that the inclusion or exclusion of these respondents would not shift any outcomes”

3) The 3rd inclusion criteria for research personnel as mentioned in the manuscript and information sheet in the supplementary data does not match with the research personnel actually involved in the study. Please specify the reason for including participants who do not fall under the mentioned inclusion criteria. [ Article line number 159,160 and 171]

We think there may be a slight misunderstanding here – if I understand correctly then what is being referred to here is that we conducted some follow-up analyses in which personnel who had vs. had not conducted human challenge trials were either included or excluded, and this varied for some of the different outcomes. However, our key eligibility criterion was to do with whether respondents had been involved in Phase III clinical trials, rather than HCTs specifically. Accordingly, there is some variation as to whether or not respondents were involved in HCTs specifically. We chose not to include those who had conducted HCTs themselves in assessments of whether they thought HCTs were ethically acceptable as these respondents would presumably be heavily biased in favor of HCTs being acceptable.

4) The study shows two different groups. It is advisable to mention them by their group name rather than respondents as it would sound less confusing for readers.

Thank you for pointing out this possible source of confusion, especially in the methods. When introducing and talking about research personnel in the methods section, we know replaced the terms ‘respondents’ with ‘research personnel’. We also made similar edits throughout the manuscript where we thought there might be confusion about who was being referred to.

5) Please mention the full form of MCMC sampling in the text. [Article line number 272]

Thank you for highlighting this – we’ve now added on Page 13 that:

“The sampling for the Bayesian regression models used Hamiltonian Monte Carlo (HMC) No-U-Turn samplings (NUTS).”

6) The total number of participants (N) should be mentioned in the table number 1 provided in the supplementary data with heading "sample information for research personnel across different outcomes" for different outcomes category.

We’ve now added a short sentence above the table indicating the total number of eligible research personnel.

Additional note: On page 22 of the manuscript, we made one additional minor change from stating that the first three factors of importance were all to do with risk to participants, to state that the first two are to do with risk, and the third relates to risk:

“Foremost among the items in importance for both the general public and research personnel were two factors that directly related to the risks posed to participants: the level of risk exposure and that participants fully understand those risks, and a third item that may be seen to relate to the risks participants face: the availability of rescue treatments.”

---

## [Editor Report · Decision Letter 1]

12 Jul 2024

Ethical Acceptability of Human Challenge Trials: Consultation with the US public and with research personnel

PONE-D-23-31643R1

Dear Dr. Elsey,

We’re pleased to inform you that your manuscript has been judged scientifically suitable for publication and will be formally accepted for publication once it meets all outstanding technical requirements.

Kind regards,

Isha Amatya, M.D.

Guest Editor

PLOS ONE
---

## [Editor Report · Acceptance letter]

21 Aug 2024

PONE-D-23-31643R1 

PLOS ONE

Dear Dr. Elsey, 

I'm pleased to inform you that your manuscript has been deemed suitable for publication in PLOS ONE. Congratulations! Your manuscript is now being handed over to our production team.

Kind regards, 

on behalf of

Dr. Isha Amatya 

%CORR_ED_EDITOR_ROLE%

PLOS ONE